# Isolation and Characterization of *Streptococcus* *mutans* Phage as a Possible Treatment Agent for Caries

**DOI:** 10.3390/v13050825

**Published:** 2021-05-02

**Authors:** Hadar Ben-Zaken, Reut Kraitman, Shunit Coppenhagen-Glazer, Leron Khalifa, Sivan Alkalay-Oren, Daniel Gelman, Gilad Ben-Gal, Nurit Beyth, Ronen Hazan

**Affiliations:** 1Department of Prosthodontics, Hadassah School of Dental Medicine, Hebrew University, Jerusalem 91120, Israel; hadar.benzaken@mail.huji.ac.il (H.B.-Z.); reut.kraitman@mail.huji.ac.il (R.K.); gilad.bengal@mail.huji.ac.il (G.B.-G.); nuritb@ekmd.huji.ac.il (N.B.); 2Institute of Biomedical and Oral Research (IBOR), Faculty of Dental Sciences, School of Dental Medicine, The Hebrew University, Jerusalem 91120, Israel; tinushy@yahoo.com (S.C.-G.); leronK@ekmd.huji.ac.il (L.K.); sivan.alkalay@mail.huji.ac.il (S.A.-O.); daniel.gelman@mail.huji.ac.il (D.G.)

**Keywords:** *S. mutans*, phage therapy, dental caries, *S. mutans* phage, bacteriophage, biofilm

## Abstract

*Streptococcus mutans* is a key bacterium in dental caries, one of the most prevalent chronic infectious diseases. Conventional treatment fails to specifically target the pathogenic bacteria, while tending to eradicate commensal bacteria. Thus, caries remains one of the most common and challenging diseases. Phage therapy, which involves the use of bacterial viruses as anti-bacterial agents, has been gaining interest worldwide. Nevertheless, to date, only a few phages have been isolated against *S. mutans*. In this study, we describe the isolation and characterization of a new *S. mutans* phage, termed SMHBZ8, from hundreds of human saliva samples that were collected, filtered, and screened. The SMHBZ8 genome was sequenced and analyzed, visualized by TEM, and its antibacterial properties were evaluated in various states. In addition, we tested the lytic efficacy of SMHBZ8 against *S. mutans* in a human cariogenic dentin model. The isolation and characterization of SMHBZ8 may be the first step towards developing a potential phage therapy for dental caries.

## 1. Introduction

Dental caries (tooth decay) is the most common infectious disease worldwide [1,2,3]. One of the main pathogens that has an important part in the development of dental caries is *Streptococcus mutans* [4,5,6]. *S. mutans* is an acidogenic and aciduric Gram-positive bacterium that naturally inhabits the oral cavity [7]. *S. mutans* is classified into serotypes c, e, f, and k, based on the chemical composition of its serotype-specific polysaccharides; 70–80% of the strains found in the oral cavity are classified as serotype c, followed by e (~20%), and f and k (less than 5% each) [8,9].

Despite the remarkable progress in reducing caries prevalence by means of fluoridation, improved oral hygiene, and increased access to dental care, dental caries remains one of the most common chronic diseases [10]. Nearly 20% of children between the ages of 2 and 4 have detectable caries and, by the age of 17, almost 80% of adolescents will have a cavity [10]. Studies concerning the global burden of this disease have revealed that untreated caries was ranked as the most prevalent health condition in the last decade, affecting the permanent dentition of 2.4 billion people [2,3]. Untreated caries can cause severe pain and mouth infection and affects not only the mastication function but also the individual’s quality of life, through such factors as smiling, speech, school attendance in children, and work productivity in adults [2,11]. 

Dental caries is directly linked with the ability of bacteria to form biofilms [12]. At this stage, the bacteria become inaccessible to antibacterial agents and the body’s immune system [13,14]. Cariogenic bacteria can readily form biofilms on the tooth surface, rapidly producing lactic acid and consequently causing dental decay [4,12,15].

At present, the incidence of caries is continuing to rise. During the past fifty years, there have not been any innovations in the prevention of dental caries. Current therapies lack sensitivity; they are not species-specific and kill pathogenic species as well as commensal species, which can protect against the formation of pathogenic biofilms. Therefore, there exists a need to re-establish and develop new therapeutic strategies that prevent or eliminate biofilm formation in more precise ways, selectively targeting cariogenic bacterial biofilms and being specifically geared towards preventing and treating dental caries in clinical practice [3,13,15,16,17,18,19,20].

A promising alternative approach is bacteriophage (phage) therapy. Phages are bacterial viruses that invade bacterial cells, disrupt their metabolism and, when in the lytic cycle, cause the bacterium to lyse [21,22,23]. Due to an increase in the prevalence of antibiotic resistance, phage therapy has gained interest in the western world [13,22,23,24,25]. The key benefits of phage therapy include: High strain specificity with low impact on the commensal microbiome; ability to multiply at the infection site and to disappear concurrently with the target pathogen; phages are natural products, devoid of apparent toxicity; they are relatively easy to isolate and to genetically engineer; phages can co-evolve with their bacterial host to kill multi-drug resistant (MDR) bacteria; and, finally, phages can efficiently destroy biofilms that conventional antibiotics usually cannot [6,13,22,23,24,25,26,27,28,29,30].

Recently, phage therapy treatments have been conducted, mainly in Emergency Investigational New Drug (eIND) applications [31], including by our group [32]. To date, to the best of our knowledge, no phage therapy treatments have been applied against caries.

Despite the established significance of *S. mutans* in dental caries and the basic understanding of the importance of phages in general, there exist very few data and studies regarding the *S. mutans* phages specifically, as well as their place regarding the environment of *S. mutans* in the oral cavity [6,13,33,34]. It should be noted that anti-*S. mutans* phage therapy is relatively difficult [33], and several attempts in the past few years have ended without success [35,36]. Thus far, only three bacteriophages infecting Mutans streptococci have been isolated from the oral cavity and their genomes have been sequenced: M102 [37], M102AD [38], and ɸAPCM01 [39].

In the present study, we describe the isolation and characterization of a new *S. mutans* phage. Additionally, its efficacy against *S. mutans*, as the first step towards the development of anti-cariogenic phage therapy, is evaluated.

## 2. Materials and Methods

### 2.1. Bacterial Growth Conditions

*S. mutans* strains were grown in brain heart infusion (BHI) broth (Difco, Detroit, MI, USA) or on BHI agar plates overnight at 37 °C in 95% air and 5% CO_2_ (*v/v*). The bacterial strains used for screening are listed in Table 1.

Ten saliva samples were collected from random healthy volunteers. The samples were streaked on mitis salivarius agar plates selective for *S. mutans* [40]. After 24–48 h of incubation at 37 °C in 95% air and 5% CO_2_ (*v/v*), single colonies were picked into BHI broth (Difco, Detroit, MI, USA) for propagation. The isolated bacteria were verified to be *S. mutans* by light microscopy (100×) and by specific *S. mutans* PCR primers (Left primer: TTGACTATTGCTGCCTTGGC, right primer: TTGTGCACTTTGAGGCGAAA, designed using Primer 3 version 4.0.0; http://primer3.ut.ee/). The oligonucleotides were confirmed by PCR on *S. mutans* strain ATCC UA159, with *E. faecalis* as a negative control. Ten strains of *S. mutans* were positively identified in this way.

The *S. mutans* strains used in this study are listed in Table 3. The ATCC strain of *S. mutans*-25175 UA159 was c serotype [41,42]. Five *S. mutans* strains were generously provided by Kyushu University in Japan, including one of **c** serotype (MT8148), two of **e** serotype (LM7 and MT703), and two **f** serotype (OMZ175 and MT6219). The serotypes of the bacteria isolated from saliva samples was determined by PCR, using primers specific to the different serotypes (c serotype primers—Left primer: TTGACTATTGCTGCCTTGGC, right primer: TTGTGCACTTTGAGGCGAAA, designed by Primer 3 version 4.0.0; http://primer3.ut.ee/) [9]. These primers target a region (1007 bp) between the DUF1700 domain-containing protein (protein ID: QIX85584.1; in *Streptococcus mutans*, CP050962.1) and a DUF4097 domain-containing protein (protein ID: QIX85585.1 in the same bacteria).

### 2.2. One Step Growth

One-step growth experiments were performed as previously described [43]. Briefly, 0.9 mL stationary-phase *S. mutans* (5 × 10^8^ CFU/mL) were mixed with SMHBZ8 for a total concentration of 1.3 × 10^6^ PFU/mL. The mixtures were incubated in 37 °C for 10 min, in order to allow phage absorption, followed by dilution by a factor of 10^4^ in fresh BHI broth and additional incubation. Then, 0.1 mL samples were retrieved from the diluted mixtures in 10 min intervals, from t = 30 to t = 100 min after phage introduction. An additional sample was taken at t = 120 min. These samples were immediately mixed with 3 mL of Soft BHI Agar (0.6% Agar) and 0.3 mL of stationary-phase bacteria, for phage enumeration by the double-layered agar plaque assay. The average phage titers at each timepoint are presented, which were used to calculate the latent time and burst size.

### 2.3. Sample Collection

Saliva samples, extracted teeth, and dental biofilm (dental plaque) from healthy volunteers were collected in the dental clinics of Hadassah Medical Center, Jerusalem (approved by the Hebrew University-Hadassah Institutional Ethics Committee, protocol no. MD-0351-16-HMO). Dental sewage was collected at the same clinic and regular sewage samples were taken from the decontamination facility in West Jerusalem.

Samples of cariogenic dentin were also collected in the dental clinics in Hadassah Ein Karem Campus of the Hebrew University in Jerusalem, from healthy volunteers who had caries lesions (approved by the Hebrew University-Hadassah Institutional Ethics Committee, protocol no. 0680-17-HMO).

### 2.4. Phage Isolation and Propagation

Two hundred and fifty-four samples of saliva, teeth, dental plaques, dental sewage, and regular sewage were tested for *S. mutans* phages. Each sample collected was centrifuged (centrifuge 5430R, rotor FA-45-24-11HS; Eppendorf, Germany) at 7800 rpm (13,500 g) for 10 min. The samples were first filtered through 0.45 µm pore-size filters (Merck Millipore Ltd., Dublin, Ireland). Then, to filter out smaller particles, 0.22 µm pore-size filters (Merck Millipore Ltd., Dublin, Ireland) were used. Samples and filtrates were kept at 4 °C.

Filtered samples were tested for lytic behavior using agarose spot testing, as previously described [44]. Briefly, 200 µL of overnight cultures of *S. mutans* (10^8^ CFU/mL) were mixed with 3.5 mL pre-warmed 0.5% agarose and then overlaid on BHI agar plates. After the agarose solidified, multiple 10 µL spots from the filtered sample were placed on it. For the negative control, BHI broth was spotted on the same plate. Plates were incubated for 24–72 h at 37 °C, under 95% air and 5% CO_2_ (*v/v*) conditions, or until plaques were observed. When detected, plaque morphologies were examined and the clearest and most lytic ones were carved out from the agar, then transferred using a sterile pipette tip into a tube of BHI broth. The phages were propagated using an overnight culture of *S. mutans* and were purified by centrifugation at 7800 rpm (13,500× *g*) for 10 min, followed by filtration through 0.22 µm pore-size filters [25]. To determine the concentration of plaque forming units (PFU), the modified double-layered agarose method was used, as previously described [44]. Briefly, phages were serially diluted by 10-fold and 5 µL drops from each dilution were spotted onto BHI agar plates, on which 0.2 mL of an overnight culture of *S. mutans* (10^8^ CFU/mL) mixed with 3.5 mL of pre-warmed 0.5% agarose (LifeGene, Israel) was spread. The plates were incubated for 24 h at 37 °C under 95% air and 5% CO_2_ (*v/v*) conditions. The number of plaques was counted the following day and the initial concentration (PFU/mL) was calculated. 

### 2.5. TEM Analysis of the Phage

To visualize the structure and morphology of the isolated phage, transmission electron microscopy (TEM) was used, following the Gill method [45]. One ml of isolated phage (10^8^ PFU/mL) was centrifuged at 19,283 rpm (centrifuge 5430R, rotor FA-45-24-11HS; Eppendorf) for 2 h. The supernatant was discarded and the pellet was suspended in 200 µL of 5 mM MgSO_4_, then incubated for 24 h at 4 °C, allowing the pellet to disperse. A total of 10 µL of the phage were mixed with 30 µL of 5 mM MgSO_4_ and spotted onto a strip of Parafilm laid on top of a paper towel. Then, 30 µL of 2% uranyl acetate was added onto each of the grids, in order to prepare them. The grids were carefully placed, using forceps, over the drops of the phage, with the carbon side facing up. After approximately 1 min, the grids were placed on drops of 2% uranyl acetate stain for another 10–15 s. After the grids dried, they were stored until future use. For the images, a transmission electron microscope (Joel, TEM 1400 plus) with a charge-coupled device camera (Gatan Orius 600) was used. Measure of virion dimensions was performed using the NIH ImageJ 2 software [46].

### 2.6. DNA Isolation and Sequencing 

The phage’s DNA isolation was performed as previously described [25], using the Norgen Biotek Phage DNA Isolation Kit (Cat. # 46800). At 37 °C, 1 mL of phage lysate (10^8^ PFU) was treated with RNase (50 mg/L) and DNase (100 mg/L) for 30 min, in order to eradicate any bacterial nucleic acids. In order to digest both phage DNase and capsid, sodium dodecyl sulfate (20%) and Proteinase K (100 mg/L) were added, and the mixture was incubated at 52 °C for 1 h. Sequencing was performed in the Core Research Facility at the Hebrew University, Hadassah Campus, as previously described [47], using a Nextera XT DNA kit (Illumina, San Diego, CA, USA) to prepare libraries. As part of it, the purification of the phage DNA was carried out, using a phage DNA isolation kit (Norgen Biotek, Thorold, Canada), AMPure XP beads, and amplification by a limited-cycle PCR. Using the Illumina MiSeq platform, the DNA libraries were tagged, pooled, and normalized in a prevalent flow cell at 2X250 base-paired-end reads. The Geneious Prime 2020.0.4 (Biomatters) software, with its plugins, was used to perform the de novo assembly with end-trimmed reads. Analysis of the open reading frames (ORFs), whole-phage genomes, phylogenetic tree generation, and comparisons to other known *S. mutans* phages were performed using the Geneious Prime 2020.0.4 (Biomatters) software and its plugins. Annotation was performed using RAST (https://rast.nmpdr.org/rast.cgi) and PROKKA version 1.12 [48]. Gene core analysis was performed using Roary, the pan-genome pipe line version 3.11.2 (22 January 2018) with its default parameters (minimum blastp percentage identity of 95) [49]. Alignment of the genome was performed using the MAFFT plugin of Geneious Prime, while the Phylogenetic tree was created using the FFT-NS-1 algorithm with a scoring matrix of 200PAM (k = 2), a gap open penalty of 1.53, and an offset value of 0.123. The phylogenetic tree was created with Geneious Prime builder using the Jukes–Cantor Genetic Distance Model and the Neighbor-Joining method with Bootstrap resampling, and was validated using the Virus Classification and Tree Building Online Resource (VICTOR of DSMZ, https://ggdc.dsmz.de/victor.php#), with both nucleotide and amino-acid modes, and VIRIDIC (Virus Intergenomic Distance Calculator, http://rhea.icbm.uni-oldenburg.de/VIRIDIC/). The phyologeny of SMHBZ8 among all sequenced phages (Appendix A) was calculated using VipTree (https://www.genome.jp/viptree/). Screening for putative virulence factors was performed using Abricate 1.01 (https://github.com/tseemann/abricate), using all of its available databases: ncbi, ecoh, plasmidfinder, ecoli_vf, card, resfinder, megares, vfdb, and argannot.

### 2.7. Determination of the Phage Lytic Activity in Planktonic Culture

Planktonic *S. mutans* growth kinetics were analyzed using a 96-well plate reader (Synergy; BioTek, Winooski, VT, USA). The plate was prepared as previously described [50], with the following modification: the outer wells of a sterile 96-well plate were filled with powder from a CO_2_ gen sachet (Thermo). To prevent the powder from spreading inside the 96-well plate, the powder-filled wells were covered with autoclave tape. The empty wells were inoculated with 160 µL of BHI, 20 µL of an overnight *S. mutans* (10^8^ CFU/mL) cultures, and 20 µL of phages at various multiplicities of infection (MOI)—0.001, 0.01, 1, 100, and 1000—in triplicate. Wells to which BHI was added (instead of the phage) served as a negative control. Before inserting the plate to the plate reader, small holes were carefully poked into the autoclave tape covering the outer wells, in order to allow the CO_2_ to diffuse out. The plate was then covered and further sealed with tape. The reader was set to record the OD of each well at 600 nm every 20 min, following 5 s of orbital shaking while incubating at 37 °C. The efficacy score was calculated using the area under the curve with the GraphPad Prism 8.02 (build 263) software, as previously described [51].

### 2.8. Determination of Phage Activity on Biofilm Formation

*S. mutans* biofilms were grown in 24-/96-well plates at 37 °C under 95% air and 5% CO_2_ (*v/v*) conditions.To determine the ability of the phage to prevent biofilm formation, an overnight *S. mutans* culture (10^8^ CFU/mL) was mixed with phages (10^8^ PFU/well) and cultivated in BHI medium supplemented with 4% sucrose for 24, 48, or 72 h.

To determine the ability of the phage to eliminate existing biofilm formations, an overnight *S. mutans* culture (10^8^ CFU/mL) was cultivated in BHI medium supplemented with 4% sucrose. After 24 or 48 h of incubation, phages were added (10^8^ PFU/well) to the wells, and incubation was continued for an additional 24 or 48 h.

The ability of phages to penetrate biofilms and their efficiency against biofilms were evaluated by several methods:

In order to quantify the biomass, crystal violet staining was used, as previously described [52]. Briefly, the wells containing the biofilm, as mentioned above, were washed with phosphate-buffered saline (PBS) and fixed with 200 µL of methanol for 20 min. For staining, 200 µL of crystal violet (1%) were added, followed by incubation for 20 min at room temperature. After the wells were washed with water, 200 µL of citric acid (37%) were added. The quantity of the biomass was read by a 96-well plate reader (Synergy; BioTek, Winooski, VT, USA) at 595 nm. Each test was performed in triplicate and the whole experiment was repeated three times.

To prepare the biofilm samples for SEM, the samples were fixed with Karnovsky’s fixative (2% PFA, 2.5% glutaraldehyde in 0.1 M cacodylate buffer, pH = 7.4) for 4 h at room temperature in a 24-well culture plate, followed by ½ diluted Karnovsky’s fixative overnight at 4 °C. Then, the biofilm samples were post-fixed in 1% O_S_O_4_ in 0.1 M cacodylate buffer for 2 h and dehydrated in a graded series of alcohols, followed by Critical Point Drying (CPD). After sputter coating with Pd/Au, biofilm samples were observed by SEM (FEI, Quanta 200; CPD—Quorum Technologies, K850 Critical Point Drier; Sputtering—Quorum Technologies, SC7620 Sputter coater). Each well was treated and visualized in duplicate.

### 2.9. Host Range Specificity Tests

The lytic activity of the phage was tested qualitatively against several *S. mutans* clinical strains, ATCC *S. mutans* strains, and other oral bacterial species from our lab collection (Table 3). For this purpose, each bacterial strain was grown overnight in their respective media, with growth conditions as shown below in Table 1.

Five hundred microliters of an overnight bacterial culture, grown as specified in Table 1, were poured onto an agar plate, until the plate was entirely covered. After allowing the plate to dry, at least 5 spots (10 µL each) of the phage sample were added. The plates were incubated at 37 °C for 24–72 h, or until plaques were visible enough to examine the differences between the strains and to test whether they were resistant or susceptible to SMHBZ8. Each experiment was repeated twice. The appearance of plaques was designated as “sensitive”, while the observation of no plaque formation was designated as “resistant”.

### 2.10. Effect of pH on Phage Activity

*S. mutans* culture (10^7^ CFU/mL) was grown overnight with phage (10^6^ PFU/mL) and 4% sucrose solution. Four test tubes were prepared in triplicate. The first contained 1 mL of an overnight *S. mutans* at 10^7^ CFU/mL and 4 mL BHI medium. The second contained 1 mL of the same bacteria, 1 mL of phages at 10^6^ PFU/mL, and 3 mL BHI medium. The third contained 1 mL bacteria, 0.5 mL sucrose (4% solution), and 3.5 mL BHI medium. Finally, the fourth contained 1 mL bacteria, 1 mL phage, 0.5 mL sucrose and 2.5 mL BHI. The tubes were incubated overnight at 37 °C in 95% air and 5% CO_2_ (*v/v*). PFU, CFU, and pH were calculated after 24 h. The pH was measured using universal pH indicator strips.

The bacteria from the third group served for the following assay, for which two tubes were prepared: The first tube consisted of 1 mL bacteria, 0.5 mL sucrose, and 3.5 mL BHI; while the second contained 1 mL bacteria, 1 mL phage, 0.5 mL of sucrose, and 2.5 mL BHI. The tubes were kept overnight at 37 °C in 95% air and 5% CO_2_ (*v/v*). PFU, CFU, and pH were calculated after 24 h.

Additionally, BHI agar plates were prepared with different pH levels (10, 9, 7, 5, and 3) using HCL and KOH solutions. Phages at 10^6^ PFU/mL were used on these plates.

After that, BHI broths were prepared at different pH levels (12, 9, 7, 5, 4, 3, and 1) using HCL and KOH solutions. Using a 96-well plate reader (Synergy; BioTek, Winooski, VT, USA), we diluted 10 µL of phage 10^6^ PFU/mL in 180 µL of each of the different BHI broths, then performed PFU on regular pH (= 7) plates of BHI.

### 2.11. SMHBZ8 Effect on Cariogenic Dentin

Seventy-six dentin samples were collected from healthy volunteers with caries lesions in the dental clinics at the Hadassah Ein Kerem Campus of the Hebrew University in Jerusalem (Figure 5). Six samples were collected from each patient (*n* = 3 test and *n* = 3 control). The dentin samples were collected with a 20102 Carisolv excavator number 4 and transferred into an Eppendorf tube containing 1.5 mL BHI. One hundred microliters of phage at 10^6^ PFU/mL were added to each test tube and 100 µL of BHI were added to the control tubes.

The cariogenic dentin sample tubes were incubated at 37 °C under 95% air and 5% CO_2_ (*v/v*) conditions. CFU/mL bacterial count was performed on BHI and mitis salivarius agar plates [40,53]. CFU/mL was assessed at 0, 12, 24, 48, and 72 h.

In addition, bacterial growth of the samples was recorded by optical density (OD) change using a 96-well plate reader (Synergy; BioTek, Winooski, VT, USA) at 600 nm. Twenty microliters of each sample were added to 3 wells with 180 µL BHI. Three wells containing 200 µL of BHI served as control. The OD was assessed at 0, 12, 24, 48, and 72 h.

The dentin samples were prepared and viewed under SEM at 0, 24, and 72 h. Sample preparation for SEM was carried out as described above.

After 72–96 h of incubation, 200 µL of the dentin samples were washed gently with PBS, then stained with a live/dead cell viability kit (Life Technologies, Waltham, MA, USA). The samples were centrifuged at 12,000 RPM for 1 min, washed with 100 µL of PBS solution, and followed by centrifugation. Then, 100 µL of the stained sample were incubated at room temperature for 20 min, centrifuged, and washed with 100 µL PBS solution, followed by another centrifugation. The stained sample was fixed with 4% paraformaldehyde for 5 min, followed by another PBS washing. The fluorescence emissions of the samples were detected using a Zeiss LSM 410 confocal laser microscope (Carl Zeiss). Red fluorescence was measured at 630 nm, while green fluorescence was measured at 520 nm. Horizontal plane optical sections were made at 5 μm intervals, from the surface outward, and the images were displayed individually. The microscopy slices were combined to form a 3D image using the Bioformats and UCSD plugins (ImageJ 1.49G) (http://imagej.nih.gov/ij/). The properties were the same for all samples. 

Three additional cariogenic dentin samples were collected from 10 patients, in order to compare between phage application and conventional antibiotics application. Ampicillin was chosen, due to the high susceptibility of *S. mutans* strains to it, as indicated by Baker et al. [54]. In order to determine the Minimal Inhibitory Concentration (MIC) for ampicillin, a stock solution of 100 µg/mL ampicillin was prepared in DDW. Serial dilutions (of 10, 1, 0.1, and 0.01 mg/mL) with DDW were made in a 96-well microtiter plate (Thermo Fisher Scientific, Roskilde, Denmark), for a final volume of 10 mL. Five hundred microliters of bacteria (10^6^ CFU/mL) were added to each tube, and the culture growth was recorded immediately after treatment for 24 h at 37 °C in a 5% CO_2_ incubator. The MIC was defined as the lowest concentration that inhibited bacterial growth. To the test tubes, we added half of the tested MIC. One hundred microliters of ampicillin were added to each of the tested samples in the antibiotics group.

The dentin samples which contained ampicillin were also analyzed by crystal violet staining, as mentioned above.

### 2.12. Statistical Analysis

The statistical significance of all experiments described here was determined using the Student’s *t*-test.

## 3. Results

### 3.1. Isolation of SMHBZ8 and Testing Its Efficiency against S. mutans

Out of 254 tested samples, only one sample resulted in an *S. mutans* phage, termed SMHBZ8. Initially, clear and small plaques on a double layer of BHI agar lawn of *S. mutans* were detected (Figure 1A). SMHBZ8 showed complete lysis of *S. mutans* following 24 h of incubation (Figure 1B). 

TEM images showed that the isometric head diameter of SMHBZ8 is approximately ~56 nm, its long non-contractile tail length was estimated to be ~244 nm long, and its tail width is ~10.9 nm (Figure 1C,D). These specified features resemble the *Siphoviridae* family of the *Caudovirales* order with B1 morphology [55,56,57]. 

Bacterial growth analysis showed the effective killing of *S. mutans* by SMHBZ8, even at an MOI of 0.1, in a dose-dependent manner, which was also reflected in the virulence index [51] (Figure 1E). Moreover, in these samples, almost no re-growth was observed for the first 70 h. The final OD reading of the MOI = 0.1 sample displayed a four-fold reduction, compared to the untreated sample. At MOI > 0.1, SMHBZ8 demonstrated almost complete inhibition of bacterial growth. SMHBZ8 MOIs of 0.1–1 inhibited the growth of the *S. mutans* culture to a lesser extent during 14–20 h, followed by rapid lysis. SMHBZ8 at an MOI of 10^2^ almost completely inhibited the growth of *S. mutans* (Figure 1E). A one-step growth experiment showed an eclipse of about 50 min, followed by a burst size of 250 (Figure 1F).

### 3.2. Whole-Genome Sequencing, and Phylogeny of SMHBZ8 

Full-genome sequencing of the phage revealed that SMHBZ8 has a genome length of 32,460 bp (Figure 2A), with a G + C content of 38.8%. 

The accession number for the genome of SMHBZ8 in the NCBI GeneBank is MT430910. DNA sequence analysis showed that SMHBZ8 appears to be closely related to other sequenced *S. mutans* phages: M102, M102AD, and APCM01 (accessions: NC_012884.1, NC_028984.1, and NC_029030.1, respectively). Comparison of the genomes of these phages (Figure 2B) using several bioinformatic tools (see Section 2) showed identity of about 75% with the other phages (Figure 2C). Gene core analysis revealed that only 7 genes were fully conserved between all the 4 phages and SMHBZ8 had 32/44 unique genes, compared to the others (Table 2). Most of the unique genes of SMHBZ8 code for hypothetical proteins. Interestingly, it has two unique lysins (ID: QKE60409.1 and QKE60410.1), which may be used in phage therapy applications. 

The comprehensive phylogeny of SMHBZ8, against all phages from RefSeq release 93, is presented in Appendix A. The exact phylogeny of this group of phages has not yet been defined by the ICTV. Nevertheless, according to the NCBI phylogeny records, they are “Viruses; Duplodnaviria; Heunggongvirae; Uroviricota; Caudoviricetes; Caudovirales; Siphoviridae” (https://www.ncbi.nlm.nih.gov/nuccore/1846462386, accessed on 9 June 2020).

Based on the other phages and reads that mapped on the terminus of the genome, we speculated that the SMHBZ8 genome is circular; however, this needs to be validated experimently. 

An virulence factor analysis performed using Abricate, release 23 (https://github.com/tseemann/abricate, accessed on 28 March 2020) on all its databases (see Section 2) did not find any known virulence factor in SMHBZ8.

### 3.3. SMHBZ8 Activity on S. mutans Biofilm Formation

SEM images showed that SMHBZ8 almost completely reduced an existing biofilm (having cultured for 24 or 48 h) within 24 h, as compared to the untreated biofilm, which looked stable and undisrupted (Figure 3A). Using crystal violet staining, we evaluated the biofilm biomass, from which we found that SMHBZ8 almost entirely inhibited the formation of biofilm for up to 72 h. The final OD reading at 24 h in the treated sample exhibited a three-fold reduction, compared to the control. Furthermore, at 48 h, the results in the treated sample displayed a 4.8-fold decrease vs. the control and, at 72 h, a 6.8-fold reduction was observed in the treated sample vs. the control (Figure 3B). In addition, a significant elimination of the existing biofilm at 24 h, 48 h, and 11 days old was observed in the treated samples: more than half the biomass was eliminated after the addition of SMHBZ8, when compared to the untreated sample, which remained stable with no reduction observed. The final OD reading of a 24-h-old biofilm treated with phage for 24 h displayed a two-fold reduction, compared to the control. For a 48-h-old biofilm which received 24 h of phage, a 3.7-fold reduction was observed, compared to the control. A 24-h-old biofilm with 48 h of phage treatment exhibited a 1.9-fold reduction vs. the control. Finally, an 11-day-old biofilm treated for 72 h with phage resulted in 1.8-fold decrease, compared to the control (Figure 3C). These results suggest that SMHBZ8 is a lytic phage which is capable of penetrating, controlling, and inhibiting the growth of *S. mutans* biofilm.

### 3.4. SMHBZ8 Range of Infectivity Tests

The range of infectivity was tested qualitatively. To this end, ten *S. mutans* clinical isolates and the three ATCC strains (Table 3) were tested for plaques, after spotting of SMHBZ8 (10^9^ PFU/mL) on the *S. mutans* lawn. All of them were found to be sensitive to the phage and plaques were visible, as demonstrated in Table 3. SMHBZ8 was able to lyse all the clinical strains tested, as well as the ATCC strains.

PCR with oligos for the different serotypes of *S. mutans* showed that the clinically isolated strains of *S. mutans* all belonged to serotype C. LM7, MT703, OMZ175, and MT6219 strains were found to be resistant to SMHBZ8 (Table 3). SMHBZ8 was found to be specifically effective against *S. mutans* serotype C. 

In addition, the specificity of SMHBZ8 was assessed against a range of 10 types of Gram-positive, Gram-negative, anaerobic, and aerobic bacteria. Table 2 demonstrates the details of the tested bacteria. Bacteria other than *S. mutans* showed complete resistance to SMHBZ8. SMHBZ8 was found to be host-specific, infecting only *S. mutans*.

### 3.5. Phage pH Properties

Caries lesions occur in the presence of sucrose and low pH; thus, we tested the phage efficacy under various pH values.

An overnight *S. mutans* growth culture had a pH of 5.1, while an overnight *S. mutans* growth culture with phage had a pH of 6. When *S. mutans* was grown overnight in the presence of sucrose, the pH was 3.5; however, when we added phage to the bacteria–sucrose solution, the pH was 6 after 24 h (Figure 4A). In the next step, when we used the *S. mutans* grown with sucrose from the previous trial (pH 3.5), we saw that the pH of the solution of *S. mutans* with sucrose was 4; however, again, the addition of the phage caused an elevation of the pH to 6 (Figure 4A).

We continued testing the phage in different pH BHI media and agar plates. In different pH BHI plates, the PFU was 10^6^ PFU/mL at pH 5 and 7. No PFU were seen at pH 3, 9, and 10 plates (Figure 4B). In different pH BHI broths, the PFU was 10^6^ PFU/mL, clearest on pH 5 plate and a bit less clear on pH 4 and 7 plates. On the pH 3 and 9 plates, the PFU was very opaque while, on pH 1 and 12 plates, no PFU were seen (Figure 4C).

### 3.6. SMHBZ8 Effect on Cariogenic Dentin

When treated with phage, up to 6 logs in CFU/mL reduction in bacterial counts was observed after 24–96 h, in comparison to the untreated samples (Figure 5A). Overall, the reduction was significant after 48, 72, and 96 h of incubation (*p* = 0.01, *p* = 0.05, and *p* = 0.001, respectively; Figure 5B). SEM images of the biofilm showed a reduction in the biofilm mass within 24 h in the phage-treated samples, compared to the control group, which looked stable and undisrupted. This phenomenon continued during the following 3 days (Figure 5C). Furthermore, CSLM showed an increase in red staining (dead cells) after 4 days in the phage-treated samples, compared to the untreated group (Figure 5D).

In both phage and antibiotics groups, we observed a decrease in bacterial growth compared to the control group; however, the reduction was more substantial in the phage group, which showed a reduction of up to 3 logs. The difference between phage and antibiotic groups was significant (*p* = 0.05; Figure 6A). In addition, the biofilm became significantly smaller in the phage-treated group (*p* = 0.04), as measured by crystal violet staining (Figure 6B). The final OD reading of 96 h was 0.3 in the phage samples, while the readings in the antibiotics and control samples were twice and four times, respectively. A difference in biofilm size and thickness was also observed in the SEM images after 96 h (Figure 6C). In addition, CSLM at 96 h indicated an increase in red staining, suggesting more dead cells in the phage-treated group (Figure 6D).

## 4. Discussion

Despite the importance of *S. mutans* in dental caries, only three *S. mutans* phage genomes have been sequenced: M102 [37], M102AD [38], and ɸAPCM01 [39]. Coinciding with previous reports [35,36], we also experienced difficulty in isolating *S. mutans* phages. In the current study, we examined 254 samples from various sources and tested these samples against 12 different strains of *S. mutans*; only then did we succeed in isolating the active *S. mutans* phage SMHBZ8. This achievement may be attributed to the large number of samples considered. Alternatively, the range of methods and the variety of processing steps could also have contributed to the successful isolation of this phage. Consequently, it can be speculated that the search and isolation of new *S. mutans* phages requires not only saliva samples, but also a broad selection of samples, such as dental plaques, teeth, and dental clinic sewage.

We characterized the *S. mutans* phage, SMHBZ8, which is a lytic phage that effectively infects and kills both planktonic and biofilm cultures of *S. mutans* in vitro, as well as in an experimental model of cariogenic dentin. Our results confirmed SMHBZ8 to be highly lytic and robust, compared to the other recently isolated phages. ɸAPCM01 was able to lyse only one strain of *S. mutans*, out of 17 that were tested [39], while M102AD was able to lyse only one strain out of 25 that were tested [38]. Only M102 succeeded to target more *S. mutans* strains, but its efficiency has never been tested against biofilms. TEM image analysis revealed that the capsid and tail measurements of the SMHBZ8 are similar to those of the other *S. mutans* phages [38,39].

According to its genome sequence, SMHBZ8 has a genome length of 32,460 bp—longer than the others: M102 is 31,147 bp in length [37], M102AD is 30,664 bp in length [38], and ɸAPCM01 is 31,075 bp in length [39]. While SMHBZ8 appears to be closely related to the other *S. mutans* phages, the differences identified in the SMHBZ8 genome, compared to the other sequenced *S. mutans* phages, suggests that it represents a new species of *S. mutans* phages that evolved, as the other phages mentioned above, independently in different regions from the same ancestor. Nevertheless, as there are only four available *S. mutans* phage genomes at present, it is hard to determine the timeline of those events.

Biofilm formations are distinct from planktonic cultures, in terms of the mode of growth. In biofilms, there is a matrix of extracellular polymeric substances (EPS) encasing the individual bacteria. Throughout the biofilm, in the interior of the micro-colonies, there is restrictive access to oxygen and nutrients and, so, the rate of cell growth may be lower; however, in the cells located on the periphery of the micro-colony, there is more access to these resources and the bacteria are more metabolically active. The process of phage infection mainly depends on their hosts, especially on their intracellular resources, which relies on their physiological status. Therefore, it is expected that phages will more effectively infect planktonic cultures than biofilms [58,59]. 

As oral biofilm elimination is one of the main targets for caries prevention, it was our main goal to test the efficiency of SMHBZ8 in eliminating *S. mutans* biofilms. As visualized in the SEM images, and further shown using CV staining (Figure 3, Figure 5 and Figure 6), SMHBZ8 effectively prevented biofilms and reduced existing biofilm. Notably, despite the differences in growth and behavior of bacteria in planktonic state versus biofilm, SMHBZ8 was found to effectively kill *S. mutans* in both types of culture. Moreover, these results display one of the most important advantages of phage therapy over conventional antibiotics: The capability to efficiently destroy the entire biofilm formation. Phage therapy provides a great complementary strategy to other applications used today in dentistry, or as a new alternative on its own.

As demonstrated in Table 3, the host range of SMHBZ8 is limited to *S. mutans*. All the tested *S. mutans* ATCC strains which demonstrated sensitivity to SMHBZ8 (UA195, 25175, 27351) belonged to the c serotype, as mentioned before (Table 3). Bacteria other than *S. mutans* showed complete resistance to SMHBZ8, thus indicating that it is, indeed, a lytic phage with a narrow host range which can only infect *S. mutans* type c. 

Few results have supported the concept of using phage therapy against caries: As demonstrated from the pH tests, when SMHBZ8 was added to a sucrose-supplemented *S. mutans* media, the media became less acidic. Additionally, SMHBZ8′s ability to kill its target bacterium was significantly diminished under a pH < 4 or pH > 9. In the future, when phages are prescribed to a patient, these results should be taken into consideration, in order to use a phage solution that will obtain the best results in the very acidic environment of oral caries lesions [60].

By using a human cariogenic dentin model, we showed that SMHBZ8 is able to reduce *S. mutans* load in cariogenic dentin. This model is very important, in a few aspects: The CFU/mL performed on BHI plates and on the mitis salivarius plates at time 0 gave almost the same values (Figure 5B,C), which strengthened the claim that *S. mutans* is, indeed, one of the main bacteria in human cariogenic lesions. In addition, there exists no other model that represents caries better, due to the difficulty in creating caries in Petri dishes and in animal models. A literature review did not reveal a proper model for cariogenic dentin yet and, as *S*. *mutans* infects the dentin, this kind of model is highly important for research purposes. Moreover, this model can help us to test more agents and materials against caries, before carrying out in vitro and in vivo models.

As mentioned before, the oral cavity is a very complicated area. The fact that the considered phage was able to reduce the bacterial load in the dentin more significantly than an antibiotic suggests that SMHBZ8 may be a possible candidate for caries phage therapy.

Future experiments using this cariogenic dentin model have been planned and involve developing phages against other bacteria in caries lesions, such as lactobacillus [61]. This will enable the creation of phage cocktails covering a range of bacteria.

To our knowledge, using *S. mutans* phages against dental caries has not yet been clinically applied [62]. Therefore, similarly to other medical fields that are now exploring the possibilities of phage therapy, dentistry can also progress to new horizons, in research as well as in therapeutics [61]. Moreover, further efforts to find and isolate more *S. mutans* phages should be carried out in the future, thus increasing the pharmacological diversity and increasing options for the creation of phage cocktails [63].

## 5. Conclusions

In conclusion, our results indicated that SMHBZ8 is a promising new lytic phage against *S. mutans*. The effectiveness of SMHBZ8 makes it suitable for future in vivo experiments involving phage therapy against *S. mutans*. Therefore, in this study, bacteriophage therapy—a new and more selective therapeutic strategy for caries prevention and management—is proposed as a potential clinical alternative. These clinical options can be applied as new alternatives to toothpaste or slow-release mouthwashes, or as a complementary topical application to existing methods.

## Figures and Tables

**Figure 1 viruses-13-00825-f001:**
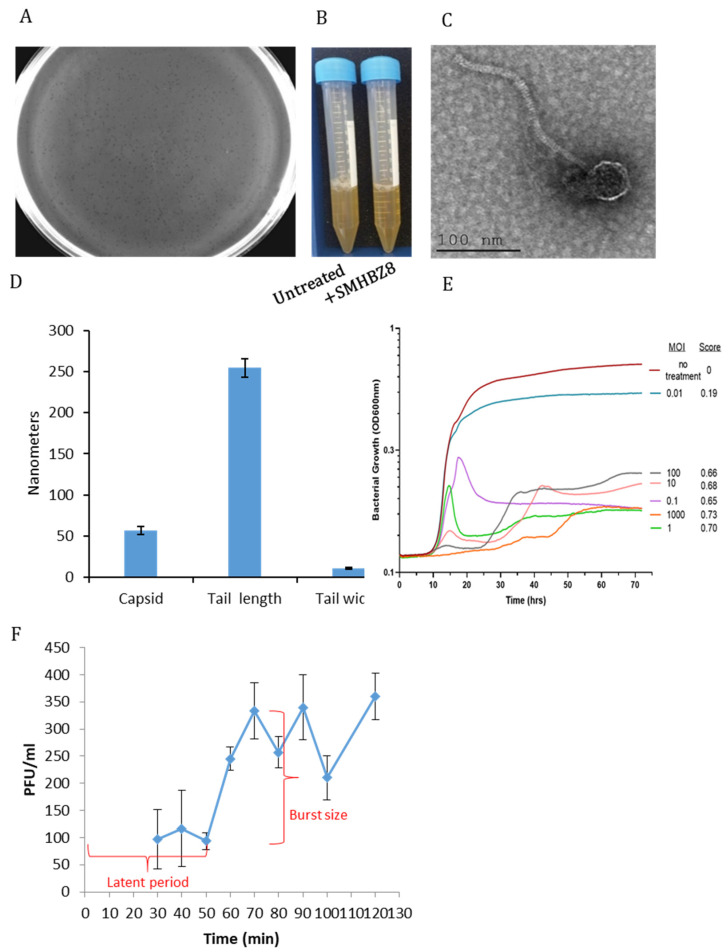
Characterization of SMBH8: (**A**) SMHBZ8 displays clear plaques on a *S. mutans* lawn after 24 h of incubation, depicting lysed bacteria. (**B**) An overnight culture of *S. mutans* incubated for 24 h in the presence of SMHBZ8 and control. Samples treated with phage were clear, as compared to the control. (**C**) TEM images of SMHBZ8 show the morphology of a Siphoviral virion head and a long tail. (**D**) Dimensions of SMHBZ8. Results are average of 4 TEM pictures. (**E**) Quantitative analysis of SMHBZ8 against a logarithmic phase *S. mutans* culture in a dose-dependent manner. The red graph shows an overnight standard growth curve of *S. mutans* in solution (i.e., the negative control). A score of the virulence index is presented as the area under the curve (auc). (**F**) A one-step growth experiment with SMBHZ8.

**Figure 2 viruses-13-00825-f002:**
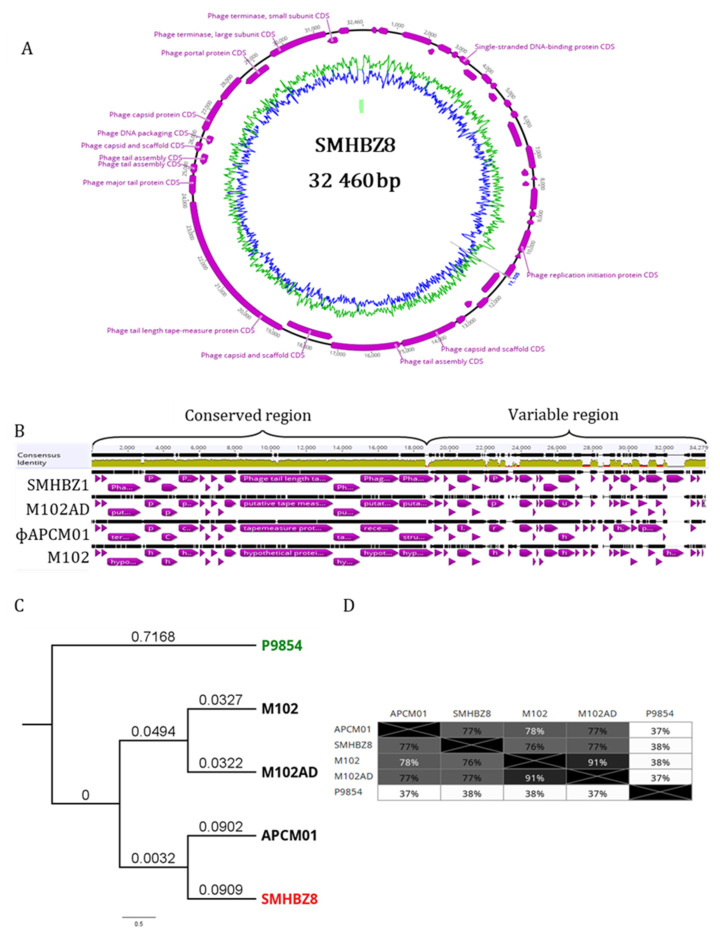
Genetic characterization of SMHBZ8: (**A**) Schematic representation of the SMHBZ8 DNA sequence. (**B**) Comparison of SMHBZ8 and three previously reported *S. mutans* phages reveal conservation in the “left” part of the genomes. The black lines show discontinuation of the genome, in comparison to the concensus of the phage genomes. (**C**) Phylogenetic tree of SMHBZ8 in relation to the other *S. mutans* phages. The numbers refer to percentage of bootstrap replicates. Phage P9854 (Accession KY705287.1) served as the outgroup. The tree was created using the Genious Prime Tree Builder software (see methods; see also Appendix A for complete phylogeny of SMHBZ8). (**D**) Percentages of nucleotide sequence identity base pairs.

**Figure 3 viruses-13-00825-f003:**
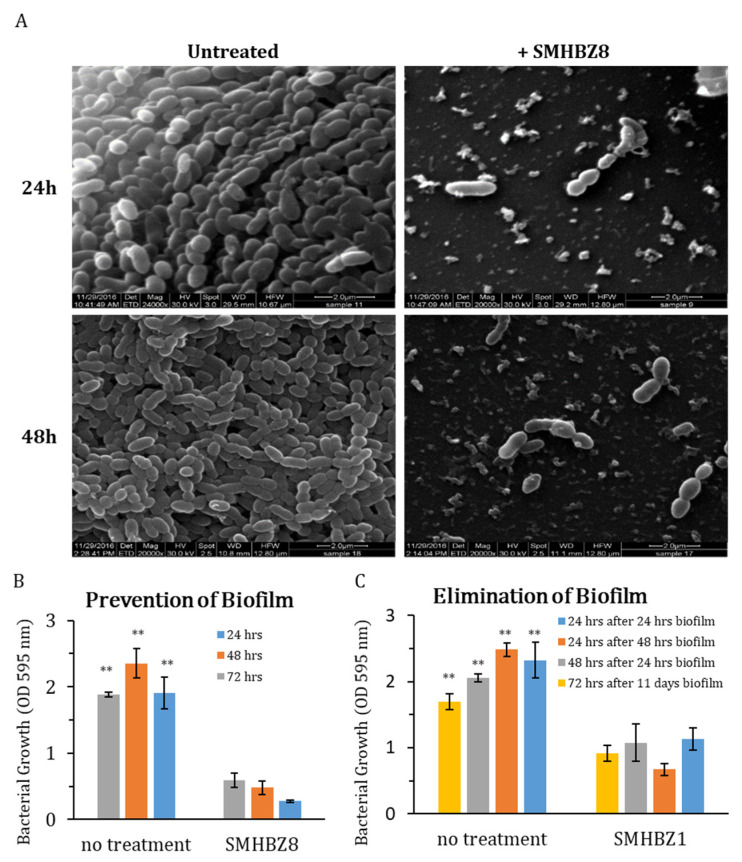
SMHBZ8 is efficient in penetrating and eliminating *S. mutans* biofilm: (**A**) SEM images show that after SMHBZ8 was added to a 24-h- or 48-h-old *S. mutans* biofilm, it was capable of eliminating it almost completely. The bacteria in the treated biofilm look scattered and disconnected, as compared to the control samples, which do not show any significant change. Using crystal violet (CV) staining: as the experiments gave similar results, a representative graph of those experiments was chosen. (**B**) This graph demonstrates there was a significant inhibition (almost entirely) of *S. mutans* biofilm formation up to 24, 48, and 72 h, as compared to the control. (**C**) SMHBZ8 was added to different *S. mutans* static biofilms at different times: For a 24-h-old biofilm, SMHBZ8 was added for either 24 or 48 h of incubation; for a 48-h-old biofilm, it was added for 24 h of incubation; and, for an 11-day-old biofilm, it was added for 72 h of incubation. In all the experiments, the pronounced reduction and destruction of mature biofilms were observed, as compared to the control experiments. These results also validate that the phage can reduce biofilm formation. All tests were performed in triplicate. Error bars represent the standard deviations. All treated samples were significantly different from the control samples (*p* < 0.01, denoted as **).

**Figure 4 viruses-13-00825-f004:**
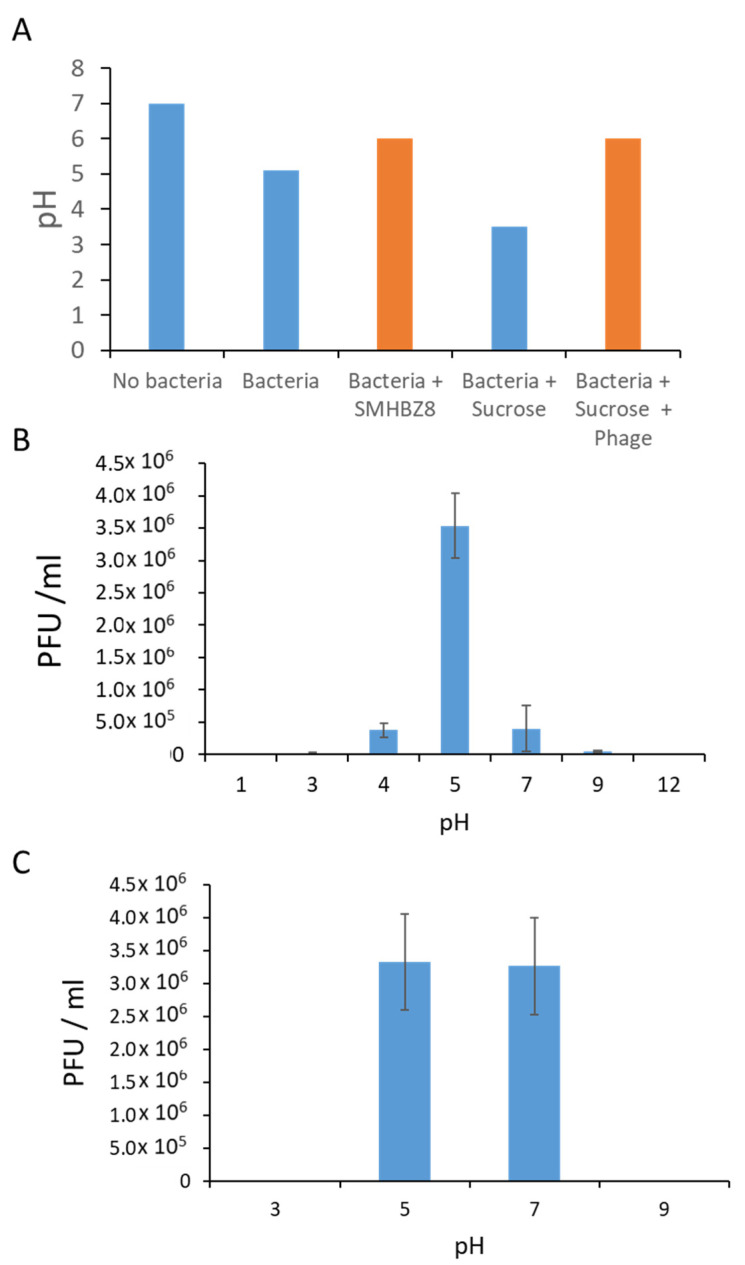
Effects of pH on SMHBZ8: (**A**) The pH in growth media of *S. mutans* in the presence or absence of sucrose and phage (orange bars). (**B**,**C**) The effects of pH on the efficacy of SMHBZ8 in liquid culture (**B**) and biofilm (**C**).

**Figure 5 viruses-13-00825-f005:**
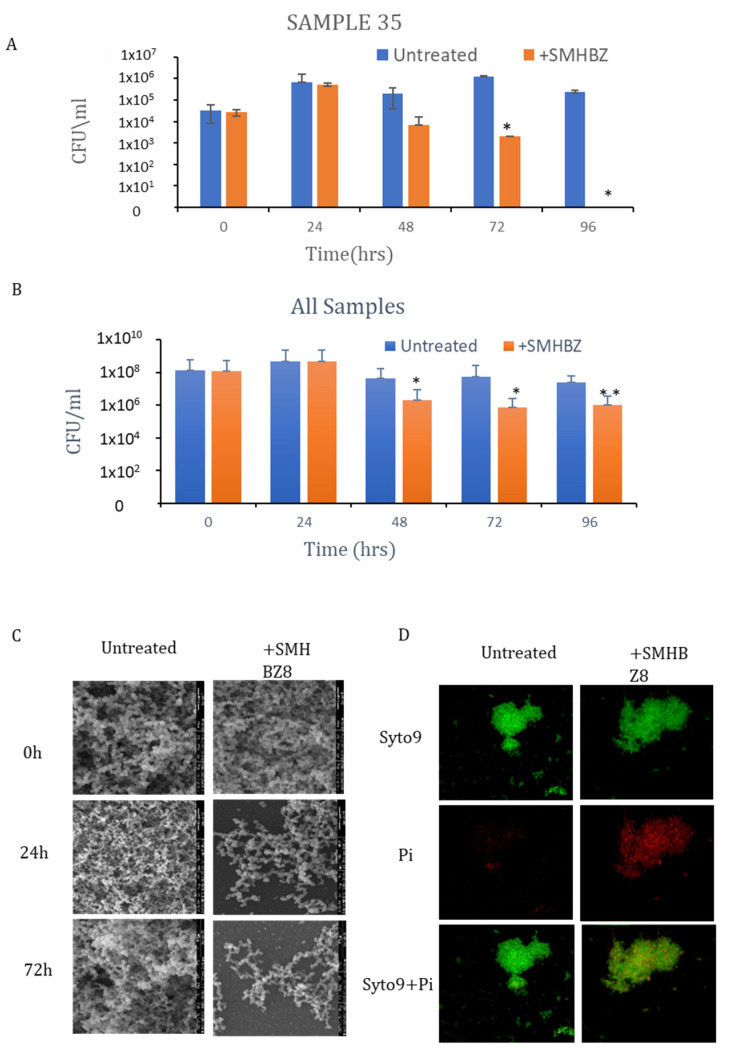
SMHBZ8 reduces bacterial load in a dentin model: (**A**) The reduction in CFU/mL of a representative dentin sample (Sample 35) treated with phages over time. (**B**) The reduction in CFU/mL in all samples on mitis salivarius plates over time. * *p* < 0.05, ** *p* < 0.005. (**C**) SEM images show that the phage reduced the biofilm mass, compared to the control. (**D**) CSLM show an increase in dead cells, as indicated by red staining.

**Figure 6 viruses-13-00825-f006:**
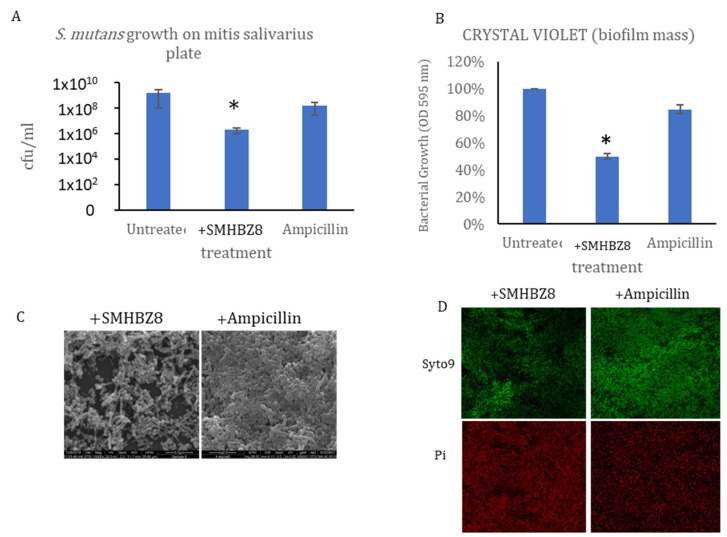
Phage vs. ampicillin: (**A**) Crystal violet staining of the biofilm, showing that phages reduced the biofilm size more significantly than ampicillin after 96 h. (**B**) CFU results after 96 h. * *p* < 0.05 (**C**) SEM pictures of phage-treated sample. The phage reduced the biofilm mass. (**D**) CSLM shows an increase in dead cells, as indicated by red staining.

**Table 1 viruses-13-00825-t001:** List of bacterial strains used in this work and their growth conditions.

Bacterial Strain	Broth	Incubation Conditions
*Streptococcus* strains		
*Streptococcus sobrinus* (lsb013)	BHI	37 °C
*Streptococcus salivarius*	BHI	37 °C
*Streptococcus gordonii*	BHI	37 °C
Other strains:		
*E. faecalis* V583	BHI	37 °C, 200 rpm shaking
*Pseudomonas aeruginosa* PA14 R	LB	37 °C, 200 rpm shaking
*Klebsiella pneumonia* (bkp016) R	LB	37 °C, 200 rpm shaking
*Actinomyces viscosus*	BHI	37 °C
*Fusobacterium nucleatum* (fs014)	LB	37 °C, anaerobic

**Table 2 viruses-13-00825-t002:** Pan-genome analysis of SMHBZ8 and its neighbor phages. Number of genes shared between the genomes. Core: Genes presented in all four phages. Soft core: Genes presented in ¾ of the phages. The full results with gene names are presented in Appendix A.

Type	Number of Genes
Core	7
Soft core including HMBZ genes	1
Soft core without HMBZ	3
Conserved in APCM01 and M102	1
Conserved in M102 and M102AD	13
SMHBZ8 unique	32
APCM01 unique	25
M102 unique	14
M102AD unique	14

**Table 3 viruses-13-00825-t003:** Bacterial strains and their sensitivity (S) or resistance (R) to SMHBZ8.

Bacterial Strain	Origin	SMHBZ1	Serotype
*S. mutans* strains			
*S. mutans* (700610)	ATCC	S	c
*S. mutans* (27351)	ATCC	S	c
*S. mutans* (ES1)	Clinically isolated from saliva	S	c
*S. mutans* (ES2)	Clinically isolated from saliva	S	c
*S. mutans* (ES3)	Clinically isolated from saliva	S	c
*S. mutans* (ES4)	Clinically isolated from saliva	S	c
*S. mutans* (ES5)	Clinically isolated from saliva	S	c
*S. mutans* (ES6)	Clinically isolated from saliva	S	c
*S. mutans* (ES7)	Clinically isolated from saliva	S	c
*S. mutans* (ES8)	Clinically isolated from saliva	S	c
*S. mutans* (ES9)	Clinically isolated from saliva	S	c
*S. mutans* (ES10)	Clinically isolated from saliva	S	c
*S. mutans* (MT8148)	Kyushu University	S	c
*S. mutans* (MT703)	Kyushu University	R	e
*S. mutans* (LM7)	Kyushu University	R	e
*S. mutans* (OMZ175)	Kyushu University	R	f
*S. mutans* (M76219)	Kyushu University	R	f
Other Streptococcus strains			
*Streptococcus sobrinus* (lsb013)	ATCC	R	
*Streptococcus salivarius*	ATCC	R	
*Streptococcus gordonii*	ATCC	R	
Other strains			
*E. faecalis* V583	ATCC	R	
*Pseudomonas aeruginosa* PA14 R	ATCC	R	
*Klebsiella pneumonia* (bkp016) R	ATCC	R	
*Actinomyces viscosus*	Clinically isolated	R	
*Fusobacterium nucleatum* (fs014)	ATCC	R	
*Staphylococcus aureus*	Clinically isolated	R	
*Escherichia coli*	ATCC	R	
*Streptococcus salivarius*	ATCC	R	
*Streptococcus gordonii*	ATCC	R	

The indication of phage sensitivity or resistance was assessed by the presence of single plaques. The results show that all the clinical strains and the three ATCC strains tested were sensitive to SMHBZ8, while all the other strains that were tested were resistant to SMHBZ8.

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
