# Peer review of "Isolation and Characterization of Streptococcus mutans Phage as a Possible Treatment Agent for Caries"

_viruses, 2021, doi:10.3390/v13050825_

Round 1
Reviewer 1 Report
The current paper decribes isolation and characterization a novel S. mutans phage.
Every researcher who has an interest in S. mutans is aware that it is hard to isolate or/and propagate S. mutans phage.
The reviewer believes that the authors made achievements.
Also, it was interesting that the authors collected 76 dentin samples for cariogenic dentin experimental model.
Minor comments
Fig 3 B & C: Error bars are differently expressed
Fig 6 A: it needs editing(horizontal axis was hidden)
Author Response
Dear editor and reviewers,
Thanks for your constructive comments. Accordingly, we did a major revision to the manuscript including new figures and tables. We also sent the manuscript for English editing in MDPI service.
Summary of major changes:
1) We also added Daniel Gelman (dnlgelman@gmail.com) who performed the one step growth experiment as an author. His affiliation is:
2 Institute of biomedical and oral research (IBOR) in the Faculty of Dental Sciences, Hebrew University-Hadassah School of Dental Medicine, Jerusalem, Israel
2) We added a one step growth experiment
3) We did a more comprehensive genome analysis using the suggested tools: roary, VICTOR VIRIDIC and VipTree. Accordingly we added supplamental table S1 and Figure S1.
4) We revised and change the figures.
5) We sent the manuscript for an English editing in MDPI service
Over all, we believe we answered each and every comment.
For more details and our point by point answers, please see below.
Reviewer #1: Comments and Suggestions for Authors
The current paper decribes isolation and characterization a novel S. mutans phage.
Every researcher who has an interest in S. mutans is aware that it is hard to isolate or/and propagate S. mutans phage.
The reviewer believes that the authors made achievements.
Also, it was interesting that the authors collected 76 dentin samples for cariogenic dentin experimental model.
> We thank the reviewer for acknowledging our achievement here.
Minor comments
Fig 3 B & C: Error bars are differently expressed
> Corrected
Fig 6 A: it needs editing(horizontal axis was hidden)
> Corrected

Reviewer 2 Report
The work describes the isolation and characterisation of a phage against Streptococcus mutans. There is some characterisation of the phage in its effectivity against a biofilms and the effect of pH on activity. The biofilm and pH aspects of the work are well done, in particular the pH aspect of the work. There are other aspects of the work could be improved. In particular some of the basic characterisation and genomic analysis.
There is no one-step growth curve presented for this new phage or alternative analysis such as virulence index. This really needs to be done
The genomic analysis also needs improvement. There has been no real contextualisation of this new phage with other phages. The comparison does not move beyond a blast comparison and some vague statements of conserved genes and variable regions. There is a plethora of software that allows core-gene analysis to be carried out between phages (Roary, core-genes, get_homologues etc). Using such software will allow homologues to be identified or also possibly extracted through Geneious software. Details of what are called “homologues” needs to be provided. This data needs to be provided and some discussion which genes are added and any putative function or that they have no function. The phylogenetic analysis as presented does not provide any context of these phages – how the current tree was produced is not clear. Standard practice for placement of phages phylogenetically requires more than one marker gene to be used (terL, polA, major capsid etc) using phylogenetic approaches. Or whole genomes using vcontact2, VICTOR, Gravity, VIPTREE etc should be used with more than the four phages included to get a better overview of how these phages are related to other phages. The phage is clearly a new phages species and probably should be stated more clearly.
Minor comments
The terminology of “anti –Streptococcus” is unnecessary and should not be used
L64 not all phage cause a bacterium to lyse. Specify lytic phages if that is the intention
L83 prefix anti , is not required here or elsewhere.
L87 phage SMHB78 ?
Figure 1 – if this is bacterial growth data, it should be a semilog plot
L92 – Figure 1 shows clear re-growth after 30 hrs for some samples, or do you mean specifically at MOI 0.1 ?
L140 need std error for measurement and from how many particles. Consider rephrasing hexagonal
L142 – Phages have not been classified on head morphology for several years, this can only be done at the genomic level now. It has a Siphovirus morphology, but it is not a Siphoviridae
L146 – Clarify what a closed circular genome is. Based on the library prep it would seem termini have not been sequenced (not suggesting this should be done ) and the circular form could be a result of a) it is a circulated permuted genome , b) assembly artefact that can occur
Was the genome checked for known lysogeny/ARGS and virulence genes ?
L151 – based on the values presented this phage most likely would not fit in the same genus as these other phages.
How many genes were shared ? how was this determined ? How many point mutations and how was this determined ?
Figure 1 D How was this constructed ? If this whole genome alignments it is not an appropriate method of phylogenetic analysis. Why are other genomes not included? There is not outgroup or support values.
The phylogenetic analysis of this phages is inadequate
L236 What titre of phage was added ? Was a dilution series carried out and the EOP determined. The current methods suggest a single titre of unknown titre was added
L244 Gram not gram
L449 - what measure is being used to claim highly lytic and robust ? There are quantitative measures such as virulence index. No such comparison seems to have been made. Some support for highly is needed . Or do they mean a broader host range . There is no one-step analysis of this phage, which would be expected for basic characterisation of phage and claims of lytic ability
L454- what do they mean by compatible ?
L458 L460 there are will defined criteria for phage species and genera. This phage is clearly a novel species based on ICTV classification of 95% ANI identity. I would suggest this is stated. Closely related is matter of debate, the other mentioned phages are the closest relative based on blast . But a more indepth analysis of these phages using one on the commonly used programmes VIRIDIC, vconctact2, VICTOR , phylogenetic analysis of 3 marker genes (terL, polA, major capsid protein etc ) as recommended by ICTV for characterisation of phage would greatly improve the genomic analysis
L461 – this statement is somewhat vague. These mechanisms of evolution could be stated for any phage system. No details of SNPs (point mutations) or the unique genes are provide or any putative function or discussion of what they might be doing .
L490 – Some acknowledgement that only a relatively small number of bacterial strains were tested here . Twenty in total, with only 4 on different serotypes.
L575 it would be easier to provide g here. The authors do provide the rotor
L576 here and elsewhere pore size filters
L581 – what titre of spots – a dilution series should really be carried out
L620 can the authors clarify the amplification – was this part of the standard NexteraXT library prep ( I am assuming so) or additional ?
L624 – version number for software and setting
L629 – what cutoffs were used in RAST ?
Author Response
Dear editor and reviewers,
Thanks for your constructive comments. Accordingly, we did a major revision to the manuscript including new figures and tables. We also sent the manuscript for English editing in MDPI service.
Summary of major changes:
1) We also added Daniel Gelman (dnlgelman@gmail.com) who performed the one step growth experiment as an author. His affiliation is:
2 Institute of biomedical and oral research (IBOR) in the Faculty of Dental Sciences, Hebrew University-Hadassah School of Dental Medicine, Jerusalem, Israel
2) We added a one step growth experiment
3) We did a more comprehensive genome analysis using the suggested tools: roary, VICTOR VIRIDIC and VipTree. Accordingly we added supplamental table S1 and Figure S1.
4) We revised and change the figures.
5) We sent the manuscript for an English editing in MDPI service
Over all, we believe we answered each and every comment.
For more details and our point by point answers, please see below .
Reviewer #2
Comments and Suggestions for Authors
The work describes the isolation and characterisation of a phage against Streptococcus mutans. There is some characterisation of the phage in its effectivity against a biofilms and the effect of pH on activity. The biofilm and pH aspects of the work are well done, in particular the pH aspect of the work. There are other aspects of the work could be improved. In particular some of the basic characterisation and genomic analysis.
There is no one-step growth curve presented for this new phage or alternative analysis such as virulence index. This really needs to be done
> We added a virulence index, using area under the curve in fig 1E and a one step growth in 1F.
The genomic analysis also needs improvement. There has been no real contextualisation of this new phage with other phages. The comparison does not move beyond a blast comparison and some vague statements of conserved genes and variable regions. There is a plethora of software that allows core-gene analysis to be carried out between phages (Roary, core-genes, get_homologues etc). Using such software will allow homologues to be identified or also possibly extracted through Geneious software. Details of what are called “homologues” needs to be provided. This data needs to be provided and some discussion which genes are added and any putative function or that they have no function.
> We performed core-genes analysis using roary and the results are presented in new table 1, table S1 and are referred in the text. See also our answer to the 5th comment of the editor above.
The phylogenetic analysis as presented does not provide any context of these phages – how the current tree was produced is not clear. Standard practice for placement of phages phylogenetically requires more than one marker gene to be used (terL, polA, major capsid etc) using phylogenetic approaches. Or whole genomes using vcontact2, VICTOR, Gravity, VIPTREE etc should be used with more than the four phages included to get a better overview of how these phages are related to other phages. The phage is clearly a new phages species and probably should be stated more clearly.
> We deeply thank the reviewer for introducing to us these great tools! Upon the request of the reviewer and the editor, we added the details of the tree parameters by Geneious Prime. We validated these result using the suggested tools VICTOR (nucleotides and amino acid modes) and VIRIDIC. In both the results matched with the Geneious results. Finally we added a comprehensive phage tree that shows the phylogeny of SMHBZ8 in between all sequenced phage using VipTree (supplemental Figure S1). The text was changed accordingly.
Minor comments
The terminology of “anti –Streptococcus” is unnecessary and should not be used
> Corrected.
L64 not all phage cause a bacterium to lyse. Specify lytic phages if that is the intention
> The sentence was change to: “Phages are bacterial viruses that invade bacterial cells, disrupt their metabolism, and when in the lytic cycle, cause the bacterium to lyse”
L83 prefix anti , is not required here or elsewhere.
> Corrected.
L87 phage SMHB78 ?
> Corrected
Figure 1 – if this is bacterial growth data, it should be a semilog plot
> Corrected
L92 – Figure 1 shows clear re-growth after 30 hrs for some samples, or do you mean specifically at MOI 0.1 ?
> By “re-grow” We mean a growth after inhibition. Sample MOI=0.01 grew from the beginning, although to a lesser extent. We clarified that in the text.
L140 need std error for measurement and from how many particles. Consider rephrasing hexagonal
> We added the measurements of the phage with error bars in figure 1D. “Hexagonal” was changed to “isometric”
L142 – Phages have not been classified on head morphology for several years, this can only be done at the genomic level now. It has a Siphovirus morphology, but it is not a Siphoviridae
> Of course, classification is done only by genomic. The TEM is just supportive visualization of that. We clarify it in the text (L138 – 144 of the revised manuscript).
L146 – Clarify what a closed circular genome is. Based on the library prep it would seem termini have not been sequenced (not suggesting this should be done ) and the circular form could be a result of a) it is a circulated permuted genome , b) assembly artefact that can occur
> We agree. We toned down the sentence and clarify it.
Was the genome checked for known lysogeny/ARGS and virulence genes ?
> We tested that using Abricate (https://github.com/tseemann/abricate) in all its databases. No problematic gene was identified. The data is presented in L.145-146 and in the Material and Methods of the revised manuscript.
L151 – based on the values presented this phage most likely would not fit in the same genus as these other phages.
How many genes were shared ? how was this determined ? How many point mutations and how was this determined ?
> As also upon the editor request we run a core analysis using roary. The results are presented now in Tables 1 and S1. Whether SMHBZ8 is a new genus, we are reluctant to decide and we trust the ICTV on this matter.
Figure 1 D How was this constructed ? If this whole genome alignments it is not an appropriate method of phylogenetic analysis. Why are other genomes not included? There is not outgroup or support values.
> As mentioned above, we revised this analysis. First, we added an outgroup and we validated the results with VICTOR (nucleotides and amino-acid) and VIRIDIC. Note that all three methods produced the same results.
The phylogenetic analysis of this phages is inadequate
> We revised it. See above and in L. 501-519 of the revised manuscript/
L236 What titre of phage was added ? Was a dilution series carried out and the EOP determined. The current methods suggest a single titre of unknown titre was added
> The titre was 109 PFU/ml (added to the revised manuscript). This assay was only qualitative assay to determine range of infectivity (Yes/No). Therefore, EOP was not done.
L244 Gram not gram
> Corrected
L449 - what measure is being used to claim highly lytic and robust ? There are quantitative measures such as virulence index. No such comparison seems to have been made. Some support for highly is needed . Or do they mean a broader host range .
> We added virulence score presented in Figure 1E based on Storms ZJ et. al, Phage. 2020;1(1):27-36.
There is no one-step analysis of this phage, which would be expected for basic characterisation of phage and claims of lytic ability
> We added one step growth (Fig 1F).
L454- what do they mean by compatible ?
> Changed to “similar”.
L458 L460 there are will defined criteria for phage species and genera. This phage is clearly a novel species based on ICTV classification of 95% ANI identity. I would suggest this is stated. Closely related is matter of debate, the other mentioned phages are the closest relative based on blast . But a more indepth analysis of these phages using one on the commonly used programmes VIRIDIC, vconctact2, VICTOR , phylogenetic analysis of 3 marker genes (terL, polA, major capsid protein etc ) as recommended by ICTV for characterisation of phage would greatly improve the genomic analysis
> As mentioned above we did a VICTOR and VIRIDIC analysis. We refer to that in the text.
L461 – this statement is somewhat vague. These mechanisms of evolution could be stated for any phage system. No details of SNPs (point mutations) or the unique genes are provide or any putative function or discussion of what they might be doing .
> The whole paragraph was rephrased to: “While SMHBZ8 appears to be closely related to the other S. mutans phages, the differences identified in the SMHBZ8 genome compared to the other sequenced S. mutans phages suggests that it represents a new species of S. mutans phages that evolved, as the other mentioned above phages, independently in different regions from the same ancestor. Nevertheless, since there are only four available S. mutans phage genomes, it is hard to determine the timeline of those events.”
L490 – Some acknowledgement that only a relatively small number of bacterial strains were tested here . Twenty in total, with only 4 on different serotypes.
> We agree with the reviewer and the sentence was removed.
L575 it would be easier to provide g here. The authors do provide the rotor
> Corrected
L576 here and elsewhere pore size filters
> Corrected
L581 – what titre of spots – a dilution series should really be carried out
> This is our standard method to isolate new phages. The first step is being carried out without dilutions. Later on we purify, re-grow and only than do dilution to have single plaques.
L620 can the authors clarify the amplification – was this part of the standard NexteraXT library prep ( I am assuming so) or additional ?
> Correct. We clarified it in the text.
L624 – version number for software and setting
> We apologize and thank the reviewer for noticing that. Currently, we do not use fastQC anf fastX in our pipeline. This sentence was omitted from the revised manuscript.
L629 – what cutoffs were used in RAST ?
> We use the online version of RAST (https://rast.nmpdr.org/rast.cgi) with its default parameters (E-value of 1-e−5.)
Round 2
Reviewer 2 Report
The authors have addressed all previous issues